# Biomarkers for Point-of-Care Diagnosis of Sepsis

**DOI:** 10.3390/mi11030286

**Published:** 2020-03-10

**Authors:** Andrew Teggert, Harish Datta, Zulfiqur Ali

**Affiliations:** 1Department of Clinical Biochemistry, James Cook University Hospital, Middlesbrough TS4 3BW, UK; h.k.datta@ncl.ac.uk; 2Institute of Cellular Medicine, The Medical School, Newcastle University, Newcastle-upon-Tyne NE2 4HH, UK; 3Healthcare Innovation Centre, School of Health and Life Sciences, Teesside University, Middlesbrough, Tees Valley TS1 3BX, UK

**Keywords:** sepsis, biomarkers, point-of-care-testing

## Abstract

Sepsis is defined as a life-threatening organ dysfunction caused by a dysregulated host response to infection. In 2017, almost 50 million cases of sepsis were recorded worldwide and 11 million sepsis-related deaths were reported. Therefore, sepsis is the focus of intense research to better understand the complexities of sepsis response, particularly the twin underlying concepts of an initial hyper-immune response and a counter-immunological state of immunosuppression triggered by an invading pathogen. Diagnosis of sepsis remains a significant challenge. Prompt diagnosis is essential so that treatment can be instigated as early as possible to ensure the best outcome, as delay in treatment is associated with higher mortality. In order to address this diagnostic problem, use of a panel of biomarkers has been proposed as, due to the complexity of the sepsis response, no single marker is sufficient. This review provides background on the current understanding of sepsis in terms of its epidemiology, the evolution of the definition of sepsis, pathobiology and diagnosis and management. Candidate biomarkers of interest and how current and developing point-of-care testing approaches could be used to measure such biomarkers is discussed.

## 1. Introduction

The World Health Organisation (WHO) has stated that the worldwide annual mortality due to sepsis is around 6 million, with the majority of these deaths being preventable [1,2]. The WHO places strong emphasis on member states’ ensuring that strategies for sepsis prevention, diagnosis and management are implemented in their healthcare systems. There is therefore an acknowledgement worldwide that sepsis is an important syndrome that causes significant morbidity and mortality, a considerable proportion of which may be preventable. In 2017, the UK Sepsis Trust suggested that the burden of sepsis on UK healthcare systems is responsible for 44,000 deaths, with an estimated 250,000 cases of sepsis annually [3]. Sepsis has become the focus of much research aiming to better understand the pathophysiological mechanisms that give rise to sepsis, particularly the apparent underpinning concept of the coupling of an initial hyper-immunological response to a counter-immunological state of immunosuppression [4], and a better understanding of the epidemiology of sepsis [5]. Another area of major interest is sepsis diagnosis, particularly the utility of biomarkers to provide more timely diagnosis; over 200 biomarkers have been investigated for this potential purpose [6].

Sepsis has been recognised for thousands of years; “sepsis” originates from an ancient Greek word which literally means “the decomposition of animal or vegetable matter in the presence of bacteria” [7]. More recently, a lay definition of sepsis was proposed which succinctly describes the condition: “Sepsis is a life-threatening condition that arises when the body’s response to an infection injures its own tissues and organs” [8]. This conveys key concepts as well as recognising severity, namely that sepsis is initiated by an invading pathogen and results in a process in which the body’s own response has a deleterious effect upon itself. This pathophysiological response can culminate in multi-organ failure, usually due to a combination of cardiovascular, cellular, coagulation and endothelial dysfunction, aptly described as “the four horsemen of the septic apocalypse” [9].

Although bacteria are recognised as the main infective agents that cause sepsis, viral and fungal infections are also responsible for a relatively small proportion of cases [10], which could account for some cases in which the underlying infective organism is not identified, despite the clinical presentation of sepsis. In the large dataset investigated by Paoli and co-workers [11], it was noted that almost 77% of cases had “unspecified septicemia”. The most frequent infective bacteria were *Escherichia coli* (7.35%), *Streptococcus* (4%), methicillin-resistant *Staphylococcus* (2.86%) and *Staphylococcus* (1.9%). It is possible that some of the unspecified septicemia cases could have been due to another infective pathogen. It has been suggested that viral sepsis should be considered in cases of sepsis with no clear bacterial infection or no other obvious causative agent [12]; however, the authors acknowledge that it is uncertain whether viral sepsis is significantly different from bacterial sepsis with regards to management, and that a diagnosis of viral sepsis may only help to direct the use of antiviral medications if deemed necessary. Fungal infections that cause sepsis are almost always contracted after a patient has been admitted to hospital and, as is the case with viral sepsis, the presentation of fungal sepsis is similar to that of bacterial sepsis [10]. However, fungi can grow rapidly and are associated with high mortality [13]. Opportunistic infection by fungal species presents an increased threat to patients who are already immunosuppressed, or to patients who are already in a critical care environment. Sepsis attributed to an infection acquired during hospital admission has been associated with a higher risk of mortality regardless of the infective pathogen, as demonstrated by Lopez-Mestanza and co-workers [14]. Their data also indicated that in this cohort of patients, just over 47% of nosocomial sepsis occurred in patients in critical care. It is possible that the severity of sepsis is influenced by both the type of pathogen and the environment in which sepsis is acquired.

Researchers have attempted to measure the financial impact of sepsis. Paoli and co-workers used a large dataset obtained from the Premier Healthcare Database in the United States [11]. The data spanned 69 months and contained 659 million patient episodes, which accounted for around 20% of hospital admissions to both academic and private hospitals. A total 2,566,689 sepsis cases were identified, the majority of which required inpatient care. The authors concluded that during 2013, sepsis accounted for 3.6% of inpatient admissions, and its treatment made up 13% of total annual hospital costs ($24 billion). This equated to just over $18,000 per admission for sepsis. Returning to the UK, it has been estimated that the cost of sepsis is around £7.76 billion per year, based on an assumed incidence of 147,000 cases [15]. While this might seem disproportionally more expensive than the American costs, the UK estimates include direct hospital costs and indirect costs, and they are therefore a reflection of the total economic impact. If only direct costs are considered (£830 million), then the cost per case falls dramatically. Differences in the models used to determine costs make direct comparisons between different healthcare systems difficult (as do the inherent differences in the ways in which the systems are financed and managed); however, it can be appreciated that financial costs incurred due to sepsis are significant.

In order to understand the importance, complexity and challenges posed by sepsis, this review provides background on the epidemiology of sepsis, outlines the evolution of the definition of sepsis and describes the postulated pathophysiological mechanisms underlying sepsis. Diagnostic and management approaches are considered, before focusing on the diagnostic potential of various biomarkers.

## 2. Sepsis Epidemiology

Despite the difficulties involved in collecting epidemiological data at national levels, work has been conducted that attempts to estimate the global burden of sepsis for both adult [13] and, more recently, neonatal and paediatric sepsis [14]. Both of these large systematic reviews focused on identifying robust epidemiological studies and performing meta-analysis to provide current best estimates of sepsis. In adults, it was estimated that there are 31.4 million cases of sepsis and 5.3 million deaths due to sepsis per year, based on a global population of 7.2 billion people [13]. In children and neonates, the annual incidence of sepsis was estimated as 1.2 million and 3 million cases respectively, based on the same global population, with combined mortality being between 11% and 19% [14]. Both studies recognised some significant limitations with their analysis. Despite wide-ranging literature searches spanning almost four decades, only 27 and 15 studies provided sufficient data to be included in the meta-analyses for the adult and paediatric reviews, respectively. Both studies also acknowledged that there were very few data available from middle- and low-income countries. Although estimates were extrapolated to the global population, it is possible that this is not a true reflection of the global burden of sepsis, as the sepsis rates in these countries are unknown. It is recognised that infectious disease rates are higher in these countries [13]; therefore, it could be postulated that the incidence of sepsis may be higher as a consequence. Although sepsis is not recognised as a distinct mortality-causing entity in the statistics compiled by the Global Burden of Disease (GBD) project (as causes of death are attributed to the initial infection [1]), recent work by Rudd and co-workers using GBD data has suggested that in 2017, the number of cases of sepsis globally was 48.9 million, with 11 million deaths as a result [16].

The UK Sepsis Trust indication of 250,000 cases of sepsis and 44,000 deaths per year would suggest a mortality rate of approximately 17.6%, but this is a very basic estimation based solely on the percentage calculated from these figures. A mortality rate of 29% was demonstrated by the ProMISe study [17], which was conducted across approximately 25% of all English NHS hospitals. The UK Sepsis Trust figures differed from those quoted by NHS England on their sepsis website entry [18], which state that 186,000 are admitted to hospital every year with sepsis, with the number of deaths quoted as 23,135 per year; the NHS figures are stated as being for the year 2015, with the source being the UK Office for National Statistics. This apparent discrepancy could be due to terminology, with the UK Sepsis Trust referring to the total number of cases, while the NHS England figures refer to those who are actually admitted to hospital. The variation in estimations of the prevalence of sepsis at both national and international levels reveals a significant issue, which is that the true burden of sepsis is not completely understood. This has been highlighted by the National Institute for Health and Care Excellence (NICE) in their research recommendations published as part of NICE guideline NG51, Sepsis: recognition, diagnosis and early management [19], which noted that epidemiological studies on sepsis within the UK were lacking, and that the number of cases and final outcomes of patients affected by sepsis could therefore not be accurately forecast.

## 3. Definition of Sepsis

The original definition of sepsis was proposed at a meeting of the American College of Chest Physicians and the Society of Critical Care Medicine in Chicago, USA during August 1991 [20]. The terms “sepsis”, “severe sepsis” and “septic shock” were considered, along with the proposal of the concept of the systemic inflammatory response syndrome (SIRS). These definitions are provided in Table 1. These definitions form stages beginning with an infection triggering the SIRS, thereby resulting in sepsis. Progress to severe sepsis through the development of hypoperfusion, hypotension or organ dysfunction is the next stage, should the patient deteriorate further. If hypotension persists even with adequate fluid resuscitation, then the patient is said to be in septic shock. These definitions were revisited at the 2001 International Sepsis Definitions Conference [21]. It was noted that the original definitions had been widely adopted, but a review was required due to the increased understanding of the pathobiology of sepsis. There was also concern that the 1992 definitions were not completely understood by clinicians, further prompting the need for review.

Despite this, there were no significant alterations of the original definitions, although the concept of SIRS was expanded to include more potential diagnostic criteria (see Table 2). There was some consideration of the use of biomarkers for diagnosis; however, there was insufficient evidence to support their use.

Between January 2014 and January 2015, the European Society of Intensive Care Medicine and the Society of Critical Care Medicine assembled the “Sepsis-3” task force with the objective of revisiting and re-evaluating the definitions of sepsis and associated terms based on emerging evidence, particularly due to progress made in elucidating its pathobiology (although it was acknowledged that full understanding of these complex processes is still far from complete) [22]. Certain key points were made, including the recognition of sepsis as a syndrome rather than as a defined condition, and the fact that there is no standardised diagnostic test to enable its detection. Crucially, sepsis was redefined as organ dysfunction due to dysregulated host response to infection. Previous concepts were challenged in light of new insights into sepsis pathobiology, with sepsis broadly described as an upregulated response to an invading pathogen mediated by endogenous processes, and involving inflammatory and anti-inflammatory pathways with alterations in many other systems within the host. The previously used SIRS criteria, of which two or more were required to be present for the identification of sepsis, were considered inadequate for the purposes of demonstrating the presence of the potentially lethal dysregulated response. Combinations of SIRS criteria occur in many patients without sepsis and, conversely, septic patients may not fulfil the SIRS criteria, leading to misdiagnosis. The Sequential Organ Failure Assessment (SOFA) score was identified as a valuable tool for identifying organ dysfunction; its use is built into the investigative pathway of suspected sepsis (see Figure 1). In addition, the Quick SOFA (qSOFA) scoring system, based on respiration rate, altered mental state and systolic blood pressure, was proposed as a simple system which can be used to easily and quickly to assess patients with suspected sepsis. With the increased emphasis placed on sepsis as a life-threatening condition, the separate term “severe sepsis” was considered to be unnecessary. Septic shock was still a recognised as a state in which hypotension becomes profound despite adequate fluid resuscitation and requires vasopressor administration. These “Sepsis-3” definitions are summarised in Table 3.

The evolution of the definition of sepsis has been driven by the increased understanding of the processes that underlie sepsis, the recognition that the previously accepted SIRS criteria were potentially misleading and the adoption of the more useful prognostic scoring systems (SOFA and qSOFA). The reduction of definitions to that of sepsis and septic shock has both simplified the definition, while focusing attention on the seriousness of these states.

## 4. Diagnosis and Management of Sepsis

The diagnosis or recognition of a patient at risk of sepsis by the use of the qSOFA and SOFA scoring systems is illustrated in Figure 1. SOFA was originally called the Sepsis-Related Organ Failure Assessment when it was proposed in 1996 [23], with the aim of providing an objective assessment of organ dysfunction using parameters that are routinely monitored. The current iteration of SOFA as presented by the Sepsis-3 taskforce is shown in Table 4.

The scoring system is simple, in that an aggregate score of two or more indicates organ dysfunction. The qSOFA is designed to be very simple; if two out of the three criteria are met (respiratory rate greater or equal to 22 breaths/minute, altered mental state or systolic blood pressure less or equal to 100 mmHg), then further investigation or monitoring, therapy escalation or critical care referral should be performed, depending on the clinical situation. Other scoring systems are used within the UK’s NHS; the National Early Warning Score 2 (NEW2) has been widely adopted, its primary aim being to enhance the identification of acutely ill patients who are clinically deteriorating. This was an update of the previous National Early Warning Score that attempted to address a number of issues, including how it could improve the identification of patients with possible sepsis. NEWS2 is designated by NHS England and NHS Improvement as the early warning scoring system that should be used throughout the NHS [24]. The use of multiple scoring systems has been called into question; Goulden and co-workers [25] indicated that NEWS is at the very least equivalent in its prognostic accuracy to qSOFA, and suggest that the use of an extra scoring system may be superfluous, as NEWS is already widely used within the NHS. It is important, therefore, that all such scoring systems are used correctly and in the most appropriate setting.

A key point of note is that there is not a recognised diagnostic test currently available for sepsis diagnosis, as stated by the Sepsis-3 task force [22]. The pathway in Figure 1 indicates that the recognition of sepsis requires careful patient monitoring and observation, with ongoing care and appropriate management. This philosophy is also found in the Sepsis Six initiative [26], which is a combination of both monitoring and proactive measures that should be performed within an hour:Deliver high-flow oxygen;Take blood cultures;Administer empiric intravenous antibiotics;Measure serum lactate and send full blood count;Start intravenous fluid resuscitation;Commence accurate urine output measurement.

These concepts were incorporated into the NICE guideline NG51 on the diagnosis and early management of sepsis [19]. Briefly, this guidance states that in cases of suspected sepsis, within an hour:The patient should be immediately reviewed and that blood-gas, glucose, lactate, blood cultures, full blood count, C-reactive protein (CRP), urea and electrolytes (including creatinine) and a clotting screen should be requested. A broad spectrum antibiotic should be administered and the patient discussed with a consultant;If indicated by lactate results or hypotension, administer an intravenous fluid bolus and refer for critical care review if necessary (lactate greater than 4mmol/L or systolic blood pressure is less than 90mmHg) to assess need for central venous access and administration of vasopressors;If the patient does not respond to treatment, then a consultant should directly review the patient.

Comprehensive, evidence-based international guidelines exist in the form of the Surviving Sepsis Campaign guidelines [27]. The Surviving Sepsis Campaign also periodically publishes sepsis bundles, which are shorter publications that aim to highlight best practice for the management of sepsis and convey this information in a way that is easy to operationalise. The most recent bundle was published in 2018 [28], with the main points being centred on measuring and monitoring lactate, taking blood cultures, administering antibiotics, administration of fluid resuscitation if hypotensive or with lactate greater or equal to 4 mmol/L, and use of vasopressors if hypotension persists after fluid resuscitation. These are broadly in consensus with the recommendations made by NICE; it could be suggested that the management approach from a guidance viewpoint is fairly standardised. The emphasis throughout is on the early recognition of possible sepsis and the initiation of prompt management and monitoring within an hour.

## 5. Pathobiology of Sepsis

The mechanisms that result in the development of sepsis are very complex and, as commented upon by the Sepsis-3 taskforce, are not completely understood [22]. However, it is recognised that sepsis is not driven purely by a host inflammatory response, but is a dysregulated host response with a combination of pro- and anti-inflammatory processes that lead to organ dysfunction [4]. There are certain well-known players in this process; cytokines are a key signalling system, the significance of which was initially highlighted by Tracey and co-workers, whose experiments outlined the potentially lethal role of tumour necrosis factor alpha (TNF-α) in experimental studies [29]. Figure 2, adapted from the works of Angus and van der Poll [30] and Wiersinga and co-workers [31] illustrates the initial events that form the pro-inflammatory response. The inflammatory response is initiated by detection of the invading pathogen. Host immune cells express pattern recognition receptors both on extracellular surfaces and in the cytosol. The extracellular receptors are mainly Toll-like receptors (TLRs), with Nod-like receptors (NODs) found intracellularly. TLRs are important for pathogen detection, recognising pattern associated molecular patterns (PAMPS) from pathogens, and recognising damage-associated molecular patterns (DAMPS) from damaged endogenous cells. The overstimulation of TLRs by DAMPS may propagate the inflammatory response in sepsis. NODs detect pathogens that invade the cytosol, leading to the formation of inflammasomes with further production of inflammatory cytokines. This promotes an inflammatory state with activation of leucocytes, complement and coagulation pathways that underpin the endothelial, cellular and cardiovascular dysfunction that characterises sepsis [9]. Although sepsis is probably most often associated with this overactive inflammatory state, the systemic dysfunction associated with sepsis cannot be solely attributed to this. Deutschman and Tracey [32] provided a detailed insight into the possible mechanisms that underpin this and suggest that disruptions in the normal homeostatic mechanisms of both the immune and neuroendocrine systems during sepsis alter cellular energy processes, disrupting endothelial and epithelial functions which can ultimately cause dysfunction at the organ level.

## 6. Biomarkers of Sepsis

In order to improve patient outcome, the Sepsis Six [26], NICE guideline NG51 [19] and 2018 Sepsis bundle [27] all state that the initial treatment and monitoring approaches should be started and completed within an hour, often referred to as the “golden hour” [33]. The instigation of treatment, including the administration of antibiotics within this first hour, has been shown to significantly reduce sepsis mortality, with decreased mean survival of 7.6% for every hour for which treatment is delayed in the following six hours [34]. The problem that arises is that even if sepsis is suspected, there are currently no biomarkers that have been completely validated that provide sufficient diagnostic accuracy to aid clinicians during this critical time period [35]. The use of biomarkers has been alluded to in the definitions of sepsis; both C-reactive protein (CRP) and procalcitonin (PCT) are inflammatory parameters included in the 2001 SIRS criteria (see Table 2). The use of PCT is described in the Surviving Sepsis Campaign guidelines [36], with the suggestion that PCT could be used for antimicrobial stewardship and for discontinuing antibiotics in patients with suspected sepsis, but who clinically have little indication of infection; however, the evidence for the use of PCT is generally of low quality and therefore these recommendations are weak. This is similar to the conclusions drawn by NICE on the use of PCT [37]; although they describe the use of PCT as promising, the final verdict is that there is not enough evidence to support its use routinely within the NHS, and they call for further research for validating its use for antibiotic stewardship in both the emergency department and critical care settings.

Perhaps the only diagnostic testing that currently has a confirmed role in sepsis diagnosis is pathogen identification by blood cultures. Obtaining blood cultures is recommended by all the previously described guidelines (it is important that this be done before the administration of antibiotics, as these can quickly sterilise blood cultures once administered [27]). These are considered to be the gold standard test for diagnosing bloodborne infections [38], despite the possible time delay (hours to days) until positive results can be confirmed. Although they can identify the responsible pathogen and direct later antimicrobial therapy, they have no real role in the initial diagnostic process when a patient first presents to the healthcare setting.

There is therefore great interest in biomarkers that could be used in either the emergency department or critical care to help effectively diagnose sepsis. A comprehensive review by Pierrakos and Vincent [39] identified 3370 studies that had investigated the use of biomarkers in relation to sepsis, with a total of 178 individual biomarkers identified. Of these, 34 had been assessed specifically for the diagnosis of sepsis. It was noted that there were a mixture of approaches, with some biomarkers being evaluated in either experimental or clinical studies, and some studies combining both approaches.

A more recent systematic review and meta-analysis was performed by Liu and co-workers [40], whose aim was to identify biomarkers that had been used to diagnose sepsis, and more specifically to differentiate between patients with sepsis and those with a systemic inflammatory response not induced by infection. They identified a total of 60 biomarkers, concentrating on the 7 for which they could find sufficient data to perform meaningful statistical analysis. The seven biomarkers identified are shown in Table 5.

They concluded that PCT, interleukin-6 (IL-6), presepsin, lipopolysaccharide-binding protein (LBP) and soluble triggering receptor expressed on myeloid cells-1 (sTREM-1) had some degree of diagnostic accuracy, with presepsin and sTREM-1 performing similarly to PCT. CD64 performed particularly well, although it was noted that the studies that tested it used flow cytometry, which may limit its use due to the need for specialised laboratory equipment.

The information provided by the previous literature searches and meta-analyses illustrates that the field of sepsis biomarker research is very active, with certain traditionally established biomarkers such as CRP and PCT being evaluated alongside newer emerging biomarkers such as CD64, presepsin and sTREM-1. In order to establish the current situation with regards to research into these biomarkers, a literature search was conducted using the PubMed database. The search terms “sepsis”, “biomarker” and “diagnosis” were used. All records from 1 January 2015 up to 2 December 2019 were searched, producing 1495 results. The titles of these were checked for suitability, and 237 articles were selected. The abstracts of these were reviewed, reducing the total number of articles for complete review to 119. These articles were selected as they specifically investigated the use of biomarker for sepsis diagnosis. Both meta-analyses and specific studies were included. For each study, the primary biomarker (or biomarkers) was identified; these usually formed the specific focus of the study. The patient group was noted (adults, paediatrics or neonates), as was the clinical area in which the study was conducted (generally the emergency department or critical care). From these 119 studies, a total of 91 biomarkers were identified (see Appendix A). The majority of these biomarkers had few references; therefore, only markers which had four or more references were selected for further investigation. A summary of the characteristics of these studies is presented in Table 6.

The biomarkers identified were similar to those identified by Liu and co-workers (see Table 5), indicating that these are still of considerable interest. Most attention has been focused on the neonatal and adult critical care populations. There has been some work conducted in the emergency department setting; it is logical to suggest that further work aimed at using biomarkers for early diagnosis should be concentrated in this particular area. The lack of studies in this area was highlighted by Shetty and co-workers, who point out that improving the diagnostic accuracy of biomarkers in the emergency department setting could help to optimise the management of patients with suspected sepsis [35].

These biomarkers are intrinsically linked with the pathobiology of sepsis, which can be illustrated by revisiting Figure 2 and expanding this to highlight the positions of the biomarkers in these pathways (see Figure 3). Some of these biomarkers are fairly well known. CRP has probably been the most widely measured acute-phase protein in routine biochemistry since its identification in 1930 [41], while PCT has been gaining momentum due to its increased production in parenchymal tissues in response to bacterial infection, and therefore as a biomarker of such [42], and for its potential use in antimicrobial stewardship [43]. IL-6 occupies a pivotal role in the inflammatory response, directly leading to production of CRP by the liver [44] as well as influencing B- and T-cell activity. The neutrophil to lymphocyte ratio has been used as an inflammatory index [45] and can be calculated from cell counts obtained from a routine full or complete blood count. Its use in the setting of sepsis has been suggested as being potentially most useful in low- or middle-income countries, as it does not require any extra specific analytical capability or cost outlay [46], although its diagnostic accuracy does seem to be limited [46,47,48] and well below that of CRP [48].

The remaining biomarkers are relatively new in comparison to the “established” biomarkers. CD64 is an IgG-binding receptor expressed by neutrophils, monocytes and macrophages [49] in response to cytokines released during bacterial infection [50]. It is bound to the cell, and the current method of choice for analysis is therefore flow cytometry. Presepsin is another biomarker that is elevated in response to bacterial infection [51]. It is a soluble fragment of CD14, a receptor that interacts with TLRs and plays a role in the recognition of PAMPS. Soluble CD14 is released, which is cleaved to form presepsin (or soluble CD14 subtype). Presepsin concentrations therefore rise when there is increased PAMPS recognition during infection. Triggering receptor expressed by myeloid cells-1 (TREM-1) is also expressed on the surface of neutrophils, monocytes and macrophages [52]. Increased expression is caused by both bacterial and fungal infection and it can bind both exogenous and endogenous ligands, which results in further upregulation of cytokine release [31]. Soluble triggering receptor expressed by myeloid cells-1 (sTREM-1) is released into the circulation, with concentrations reflecting the cellular expression of TREM-1 [52]. Soluble urokinase plasminogen activator receptor (suPAR) is also released from activated cells during the immune response (similarly to sTREM-1), with increased levels of activation resulting in higher serum concentrations [53]. The findings for these biomarkers are summarised in the following subsections for each of the populations and settings studied.

### 6.1. Procalcitonin

#### 6.1.1. Neonates

The largest dataset was presented by Pontrelli and co-workers, who conducted a meta-analysis of 17 studies [54]. This included neonates and paediatrics; however, they were unable to perform any meaningful statistical analysis on the paediatric group due to the lack of study data. In the neonatal group, they suggested that PCT has moderate diagnostic accuracy, but further studies are required. A more recent study that investigated the use of presepin and PCT for the diagnosis of early neonatal sepsis also suggested that PCT performed relatively poorly, calculating the area under the curve (AUC) for PCT as 0.599 [55]. This study did have some limitations in that the study group was fairly small (51 neonates) and there was no defined control group. Other studies have demonstrated a significant difference in PCT concentrations between neonates with sepsis and healthy controls [56], and that PCT is higher in proven as opposed to probable sepsis; it is also higher in non-survivors than survivors, hinting at possible prognostic use [57]. Very encouraging diagnostic performance was found by Hahn and co-workers, who calculated an AUC of 0.87 for PCT in differentiating between controls and cases of proven sepsis [58]. It should be noted that CRP used for the same purpose had an AUC of 0.96, suggesting that PCT was outperformed by CRP. Interestingly they also suggested that using a PCT/CRP ratio has a potential role in differentiating blood-culture proven sepsis from cases of suspected sepsis; this outperformed both PCT and CRP for this particular purpose. Combining PCT and CRP was also explored in an earlier study, which suggested that doing so increased the diagnostic accuracy beyond that of each individual biomarker [59]. In this study, the AUCs for PCT, CRP and PCT and CRP combined were 0.811, 0.701 and 0.813 respectively; the increase was however very modest. A more recent study which also suggested combing PCT with other biomarkers was conducted by Ahmed and co-workers [60]. They measured several biomarkers including PCT, presepsin, interleukin-6 and interleukin-8 in an attempt to diagnose early-onset neonatal sepsis. Presepsin outperformed PCT (AUCs 0.934 vs. 0.798 respectively); however, the authors suggested that a combination of these markers with the inflammatory cytokines also studied should be used. Unfortunately there was no further data analysis performed on the combination of these markers to support this. The size of the study was very small, with the experimental group comprising 30 neonates with suspected sepsis and a control group of 30 matched neonates, although the authors did acknowledge this as a potential limitation of their study.

#### 6.1.2. Adults, Intensive Care Units (ICU)

Adult critical care patients is the most studied group for measurement of PCT, and its performance in this patient group is apparently generally very good; several studies have suggested that the diagnostic accuracy of PCT used to distinguish between controls and septic patients is very high, with AUCs of 0.99 [61], 0.981 [62], 0.972 [63] and 0.92 [64] reported. However, these results must be treated with some degree of caution as most have been obtained in studies with relatively low numbers of participants; the AUC of 0.99 obtained by Mustafić and co-workers [61] was based on a group of 52 patients, although they did acknowledge this as a potential limitation of their work. A recent meta-analysis by Kondo and co-workers [65] which reviewed the findings of 19 studies containing a total of 3012 patients calculated an AUC of 0.84 for PCT and 0.87 for presepsin. Although promising, they suggested that the diagnostic performance of PCT (and presepsin) is useful, but that they should be used in combination with other diagnostic tools, as the quality of evidence they encountered was considered to be low. PCT does appear to have an advantage when used in populations acute kidney injury (AKI), with Nakamura and co-workers [66] demonstrating that PCT is more accurate than presepsin for diagnosing sepsis in AKI patients. Other studies have indicated a more modest performance when using PCT for sepsis diagnosis, with AUCs ranging from 0.7 to 0.9 [67,68,69]. There is also conflicting evidence on the possible correlation of PCT with SOFA for predicting organ dysfunction. Although Mustafić and co-workers [61] concluded that PCT was a good marker of organ dysfunction, other studies have disputed this, suggesting that SOFA actually outperforms PCT for the prediction of sepsis [70]. PCT does seem to have some use in predicting patients who are eventually diagnosed with bacteraemia [71,72], which is perhaps not surprising considering the mechanism by which PCT rises in response to bacterial infection. A further study that included adult patients in various medical wards also reported similar findings [73], which suggests that this predictive ability is not restricted to the critical care setting. The use of PCT as a prognostic tool has been questioned, with two studies suggesting that PCT levels do not differ significantly between survivors and non-survivors of sepsis [74,75]. The role of PCT may therefore be more suited to the initial diagnosis of sepsis, rather than for any prognostic modelling in adults.

#### 6.1.3. Adults, Emergency Department

Only two studies were found that used PCT in the emergency department; however, both studies used the more recent Sepsis-3 definitions as opposed to the older SIRS criteria. Both studies were relatively large, with several hundred patients enrolled, and both demonstrated lower diagnostic accuracy compared to studies in the critical care environment. In a total cohort of 1572 patients with suspected sepsis, the diagnostic accuracy of PCT was found to be 0.68 [76]. The authors considered other biomarkers in combination and also determined performance based on the older Sepsis-2 criteria. They found that all biomarkers performed better when using the Sepsis-3 criteria, and also indicated that combining biomarkers elevates the diagnostic accuracy above that of any individual marker. Better performance was demonstrated by Kim and co-workers [77], who calculated the diagnostic accuracy of PCT for both sepsis (470 patients) and septic shock (109 patients) and obtained values of 0.745 and 0.784, respectively.

It is worth noting that the larger studies have indicated lower diagnostic accuracy for PCT when compared to studies with fewer patients. Additionally, the smaller studies are generally conducted in the critical care environment, where a possible selection bias exists as patients are very ill; a large-scale study in this environment could provide some very useful data.

### 6.2. Presepsin

#### 6.2.1. Neonates

The largest study was a meta-analysis conducted by Bellos and co-workers [78]. They identified 11 studies comprising a total number of 783 neonates and calculated a pooled AUC of 0.9751 for the use of presepsin to diagnose sepsis. This suggests very high diagnostic performance although, as a slight caveat, they described presepsin as a candidate biomarker for sepsis diagnosis, as further work is required to define specific cut-off values for both preterm and term neonates. It is interesting to note that they also calculated AUCs for CRP (0.8580) and PCT (0.7831), indicating superior performance of presepsin over both (and of CRP over PCT). Similarly high performance of presepsin was demonstrated in individual studies by Montaldo and co-workers [79], with an AUC of 0.97, and Xian and colleagues [80], who reported an AUC of 0.942. A recent study by Deǧirmencioǧlu and co-workers [81] calculated an AUC of 0.939 for the diagnosis of late-onset sepsis in preterm neonates, which was slightly outperformed by IL-6 (AUC of 0.959). This was in contrast with the findings of Ahmed and co-workers [60] who found that presepsin was superior to IL-6 (AUCs of 0.934 and 0.751, respectively), although their study was focused on the diagnosis of early-onset sepsis. It is possible that early- and late-onset sepsis should be considered separate entities when investigating possible diagnostic approaches. The differences between early- and late-onset sepsis and the use of presepsin was investigated in a meta-analysis of 10 studies by van Maldeghem and co-workers [82], who calculated an AUC of 0.9412 for diagnosis in early-onset sepsis. Unfortunately they were unable to calculate an AUC for late-onset sepsis due to a lack of data on this population, although by combining all data they were able to calculate an overall AUC 0f 0.9639 for neonatal sepsis as a whole. The difference in AUCs for all neonates combined and the early-onset sepsis population highlights the potential need for appropriate diagnostic approaches depending on the clinical scenario. It should again be recognised that the patient cohorts in these studies were fairly small (fewer than 50 in most) which must be considered as a limiting factor. A similarly sized study reported a lower AUC of 0.758 for presepsin, while also reporting an AUC of 0.599 for PCT [55]. The value for PCT was also lower than has generally been reported in other studies considered previously. The authors also combined presepsin and PCT and actually found that this decreased the diagnostic power, producing an AUC of 0.263; this finding was not commented upon. Despite this, the combination of presepsin with other biomarkers such as CRP, PCT or IL-6 has been said to increase overall diagnostic power [57]. One study based on a cohort of preterm neonates born before 34 weeks gestation [83] measured presepsin at enrolment, and after three and seven days. The initial presepsin measurement was used to calculate an AUC of 0.864 for sepsis diagnosis. They also noted a decline in later presepsin concentrations and suggested that this might be an indicator of better outcomes. However it has been demonstrated separately that a decline in presepsin after birth is a normal physiological event in both neonates delivered normally and those delivered by caesarian section [84]. This is an important finding and would need to be considered if presepsin were to be used for sepsis diagnosis in neonates.

The measurement of presepsin in older children and adolescents was considered in a systematic review by Yoon and co-workers [85]. After conducting a wide-ranging literature search that identified 129 studies, their final meta-analysis was limited to just four studies of 308 patients aged between one month and 18 years of age. Despite this, they were still able to conclude that the measurement of presepsin is potentially useful for diagnosis of paediatric sepsis. They calculated an AUC of 0.925 for presepsin, which was superior to that of both PCT and CRP (AUCs of 0.820 and 0.715 respectively), although the authors clearly recognised that these results should be interpreted with a degree of caution due to the heterogeneity and small number of studies included in their analysis.

#### 6.2.2. Adults, ICU

Most studies in the adult critical care setting have focused on comparing presepsin with PCT. Generally, the performance of presepsin has been described as comparable to that of PCT, with no real advantage gained from using presepsin as a separate, stand-alone biomarker [71,86,87]. This has also been demonstrated in various medical wards outside of the critical care environment [72]. Another study suggested that PCT outperforms presepsin, reporting AUCs of 0.791 and 0.674, respectively [88]. A further point of note from this study is that CRP outperformed both, with a relatively high AUC of 0.903. The apparently relatively equivalent performance of both presepsin and PCT was described in a meta-analysis by Kondo and co-workers [65], which produced AUCs of 0.87 and 0.84, respectively. They concluded that either biomarker could be used in combination with other biomarkers as a potential diagnostic approach. It is interesting to note that the 19 studies they reviewed were all observational; no randomised controlled trials were found, which resulted in the evidence being considered of low quality. Miyoshi and co-workers demonstrated that the concentration of presepsin is influenced by renal function, and that there is a correlation between declining renal function and increasing presepsin concentration [89]. Although this study was conducted in patients without any evidence of bacterial infection or sepsis, similar results have been demonstrated in septic patients with and without AKI [66], leading to the conclusion that PCT has much better diagnostic accuracy in patients with AKI than presepsin. An increase in presepsin was also noted in AKI patients without sepsis, in line with the finding of Miyoshi and co-workers [89]. It is important to ensure that the correct analytical target is measured; one study described the use of presepsin, but closer inspection revealed that they had actually measured soluble CD14 and not presepsin (soluble CD14 subtype) [90].

#### 6.2.3. Adults, ED

Only two studies utilised presepsin in the emergency department setting. The first was larger, with a cohort of 223 patients with suspected sepsis as defined by the Sepsis-3 criteria [91]. Although presepin demonstrated fairly good diagnostic accuracy, with an AUC of 0.775, it was outperformed by PCT (AUC of 0.815). The authors concluded that the introduction of presepsin is difficult to justify in the emergency department setting due to the better performance of PCT. A smaller study with 72 sepsis patients described better performance for presepsin versus PCT, with AUCs of 0.975 and 0.874, respectively [92]; again, it must be noted that smaller studies tend to produce apparently better diagnostic performance.

### 6.3. CD64

#### 6.3.1. Neonates

Two meta-analyses of the use of CD64 in neonates were published relatively close together, but with slightly conflicting conclusions. Shi and co-workers [93] included 17 studies with a total of 3478 patients and calculated a pooled AUC of 0.866, but despite this, they concluded that specificity and sensitivity was relatively low and was outperformed by PCT. They did suggest that combining CD64 with other biomarkers such as PCT could increase the overall diagnostic accuracy, and that the use of CD64 as an individual biomarker should be treated cautiously. Dai and co-workers [50] included 7 studies with a total of 2213 patients. Their pooled AUC was calculated as 0.88—very similar to that of Shi and co-workers [93]. However, their sensitivity and specificity was similar to that of PCT, and they suggested that CD64 could be reliably used to diagnose sepsis in neonates. Smaller studies again demonstrated potentially exaggerated diagnostic performance; a study that compared 30 controls to 60 septic neonates indicated that CD64 had both negative and positive predictive values of 100%, and although receiver operator characteristics were calculated, the AUC was not provided [94]. Such claims in relatively small studies must be treated with caution. Other studies have indicated that while CD64 may be a useful biomarker for sepsis diagnosis, it is best combined with other biomarkers such as CRP, PCT or sTREM-1 to increase overall diagnostic power [59,95].

#### 6.3.2. Adults, ICU

A single meta-analysis was found that considered 14 studies with a total of 2471 patients [96]. The calculated pooled AUC for diagnosing sepsis was 0.94, with very good sensitivity and specificity. The authors concluded that CD64 was superior to CRP and PCT (the AUC for both was calculated as 0.84). It should be noted that the values obtained for CRP and PCT from this analysis were still very good, and would be useful if they could be consistently achieved. Very good diagnostic performance was also demonstrated in other studies, with AUCs of 0.99 [64] and 0.928 [97]. The limitations of these studies were their very small study cohorts (27 septic patients versus controls and 40 septic patients versus controls, respectively), which calls into question the high diagnostic power described. Combining CD64 with other biomarkers (such as CRP and PCT) to increase diagnostic power has also been suggested in adult patients [98] and its use as a prognostic tool has been investigated, although there does not appear to be any difference in the kinetics of CD64 between survivors and non-survivors when assessed through serial measurements [75].

#### 6.3.3. Adults, ED

Only two, small studies were found that used CD64 in an emergency department setting. Tan and co-workers [99] enrolled 51 patients, with sepsis confirmed in 42. In this group, CD64 was calculated to have an AUC of 0.88, and the authors suggested that it could be used for screening for bacterial infections. Recent work by Ghonge and co-workers [100] measured CD64 in 37 specimens from eight patients using a novel smartphone-based technique. Although this study was not a specific clinical trial, it did demonstrate the potential of smartphone imaging coupled with microfluidic biochip technology to provide an alternative to the traditional laboratory-based flow cytometry methodology for analysis.

### 6.4. IL-6

#### 6.4.1. Neonates

A study by He and co-workers [101] investigated several biomarkers for the identification of neonates with early-onset sepsis. Their patient cohort contained 151 neonates, 61 of whom had proven or probable sepsis, with the remaining 83 classified as non-infected. IL-6 was said to be useful in predicting sepsis, with an AUC of 0.706. PCT and CRP performed slightly better, with very similar AUCs calculated (0.723 and 0.720, respectively). These findings were supported by an earlier study, which indicated that there was a significant difference in IL-6, CRP and PCT concentrations between control and septic patients [56]. The authors also looked at cases of early- and late-onset sepsis and suggested that there was a significant difference in the concentrations of IL-6 and CRP between these two groups. They further suggested that combining IL-6 and CRP increased the sensitivity and specificity when compared to the use of either biomarker individually. This combination was also advocated by Rashwan and co-workers [57], who concluded that the use of individual biomarkers is not suitable for sepsis diagnosis. Despite this, they concluded that IL-6 has similar diagnostic power to presepsin, although both were outperformed by CRP. Combining IL-6 with other biomarkers such as PCT or presepsin was more recently suggested by Ahmed and co-workers [60], although their work suggests that IL-6 was outperformed by both presepsin and PCT. However, this was a small study, with 30 neonates with suspected early-onset sepsis (of which only 13 had positive blood culture results) and 30 controls. The largest analysis was performed by Sun and co-workers [102], who conducted a systematic review that identified 31 studies, comprising 1448 septic neonates and 1628 non-septic neonates. They concluded that IL-6 has relatively high diagnostic accuracy for identification of early-onset sepsis, with an AUC of 0.92. They considered the quality of evidence to be moderate and suggested that IL-6 should be used in conjunction with other diagnostic approaches.

#### 6.4.2. Adults, ICU

Two studies investigated the use of IL-6 along with PCT and CRP for the diagnosis of sepsis [62,103]. Both reported fairly good diagnostic performance for IL-6, with relatively similar performance for PCT. Gao and co-workers reported an AUC for CRP of 0.988, higher than that of the IL-6, which they failed to comment upon. They also measured the biomarker decoy receptor 3 (DcR3), which was the focus of their work. The AUC of DcR3 was 0.989, almost identical to that of CRP; however, despite this, DcR3 was said to be diagnostically superior, especially when combined with PCT. The apparent disregard of CRP must be questioned; if CRP had similar diagnostic power to DcR3, then why was that not considered in combination with other biomarkers? A later study suggested that IL-6 may possibly have a role in sepsis diagnosis [70], as they noted elevated IL-6 concentrations in septic patients compared to controls. IL-6 correlated with SOFA, although it did not appear to be useful for predicting mortality. An early study also demonstrated a relationship between IL-6 and SOFA, but again, the use of IL-6 in predicting outcome was limited [89]. The most comprehensive review of the diagnostic potential of IL-6 was conducted by Molano Franco and co-workers [104] as part of the Cochrane Database of Systematic Reviews, which was based on 23 studies containing 4192 patients. They concluded that further studies are required to thoroughly investigate the diagnostic potential of IL-6 due to the heterogeneity present in the existing studies, which prevented them from calculating diagnostic accuracy estimates. They did, however, identify 20 studies that are yet to be formally published, which may alter their final conclusions when available.

### 6.5. sTREM-1

#### 6.5.1. Neonates

A systematic review identified nine studies containing 961 patients [105]. Due the difference in study designs and populations (four studies in neonates, two in paediatrics and two in paediatrics specifically with neutropaenia) meta-analysis of the data was not possible; some of the data indicated a potential role for sTREM-1 as a biomarker of sepsis, although no strong conclusions could be drawn. Combining sTREM-1 with PCT, CRP or IL-6 was suggested to increase diagnostic power. One study did claim that sTREM-1 had sufficient power to be used as a stand-alone biomarker [95]; however, this was a very small study comparing 30 cases of sepsis to 30 controls, so this conclusion is questionable.

#### 6.5.2. Adults, ICU

Comparison of sTREM-1 to PCT and CRP for diagnosis of sepsis has been performed, with results suggesting that sTREM-1 may have better performance (AUCs 0.82, 0.77 and 0.72 for sTREM-1, PCT and CRP, respectively) [106]. The authors also hinted at a possible prognostic role for sTREM-1, noting that concentrations were significantly higher in non-survivors compared to survivors, which was corroborated by other work [75,90,103]. In addition to its predictive value, its potential superiority to PCT and CRP was demonstrated by Brenner and co-workers [103]. They calculated AUCs for sTREM-1, PCT and CRP of 0.955, 0.844 and 0.791, respectively, although the study was fairly small, with a sepsis group of 60 patients compared to a control group of 30. However, a study conducted by Jedynak and co-workers [107] suggested that sTREM-1 was outperformed by both CRP and IL-6 for the diagnosis of severe sepsis and septic shock, although sTREM-1 did outperform PCT for the diagnosis of severe sepsis. Interestingly, the authors suggested that none of these biomarkers could be used to diagnose sepsis in the early stage, when attempting to differentiate between sepsis and SIRS. This again demonstrates the difference in diagnostic potential depending on which definition of sepsis is used.

### 6.6. suPAR

No specific neonatal or paediatric studies were identified. One large systematic review and meta-analysis investigated the measurement of suPAR in adult patients in both the ED and ICU settings [108], both for diagnosis of sepsis and as a prognostic marker. They identified 30 studies that included a total of 6906 patients. Additionally, 17 studies comprising 2722 patients investigated the use of suPAR for diagnosis, revealing an AUC of 0.83, which they described as moderate diagnostic performance. Nineteen studies, which encompassed a larger number of patients (5622), considered the prognostic power of suPAR and calculated the AUC as 0.78, suggesting that higher suPAR concentrations indicate potentially poorer patient outcomes. The authors noted that suPAR had higher specificity than either PCT or CRP, meaning that it may be useful in ruling out sepsis in suspected cases; its use in combination with other biomarkers could therefore provide a more powerful diagnostic approach.

All other suPAR studies were conducted on adult ICU patients. Two relatively small studies designed to test the diagnostic power of suPAR demonstrated very high performance, with AUCs of 0.99 [109] and 0.938 [63]. However, the other biomarkers considered in these studies also demonstrated similarly high diagnostic performance (DcR3, PCT and lactate, with AUCs of 0.99 [63], 0.972 [63] and 0.84 [109], respectively). This high level of performance is questionable due to the small size of these studies (40 or fewer patients in each group), although the authors of both studies did recognise this as a limitation of their work. An earlier study by Zeng and co-workers [67] measured suPAR, PCT and CRP in patients with sepsis and SIRS and a healthy control group. They found that suPAR outperformed CRP for the differentiation of sepsis and SIRS, but was itself outperformed by PCT (AUCs for suPAR, CRP and PCT were 0.817, 0.681 and 0.892, respectively). By combining suPAR and PCT, they achieved an AUC of 0.927. This supports the later work by Huang and co-workers [108] who suggested the combination of suPAR with other biomarkers as a potential diagnostic strategy.

### 6.7. Biomarker Diagnostic Cut-Off Values

Despite the apparently encouraging diagnostic performance exhibited by these biomarkers, comparisons between individual studies can be confounded by the varying diagnostic cut-off values for specific biomarkers and the various analytical methods used. This has been highlighted by the findings of the meta-analyses described previously. Of the currently reviewed biomarkers, only PCT has arguably recognised reference limits. These are based on work by Harbarth and co-workers [110] and are summarised by Thermo Fisher in relation to their Brahms PCT assay [111] as an increase in PCT concentration that mirrors the severity of disease, with higher concentrations associated with sepsis. Despite this, the meta-analysis by Kondo and co-workers [65] highlighted the lack of any pre-specified diagnostic cut-offs in all studies reviewed with a range of PCT values from 0.28 to 4.4ng/mL noted. Pontrelli and co-workers [54] also noted large variation in cut-off values in their meta-analysis of neonates, with values from 0.5 to 25ng/mL noted. They suggested using 2 to 2.5ng/mL in neonates, as this was proposed by the European Medicines Agency Expert Meeting on Neonatal and Paediatric Sepsis [112], although it was recognised that the available evidence upon which this recommendation was made is controversial. The lack of a pre-specified diagnostic cut-off is also apparent with presepsin, with both the meta-analyses led by Bellos [78] and Kondo [65] in neonatal and adult populations, respectively, limited due to many studies lacking such pre-specified cut-off values, which increases the risk of bias in these studies. In neonatal populations, Bellos and co-workers [78] stratified cut-off values into ≤650, 650 to 850 and ≥850pg/mL, while van Maldeghem and co-workers [82] noted a range of between 305 and 885pg/mL in the studies analysed. In older children, Yoon and co-workers [85] noted a range of cut-offs from 240 to 1014pg/mL, with higher diagnostic accuracy achieved at values greater than 650pg/mL. However, as the studies had considerable heterogeneity and low sample numbers, they concluded that it would be inappropriate to suggest that this could be implemented clinically.

CD64 currently has no recognised diagnostic cut-off value, and many studies are hampered by the lack of any pre-specified diagnostic cut-off [96]. Dai and co-workers [50] noted a range of diagnostic cut-off values, which was apparent even when different studies used the same analytical methods, again highlighting the difficulty in comparing study results. Interestingly, despite both the differences in analytical methods and differing diagnostic cut-off values, the meta-analyses by both Shi and co-workers [93] and Dai and co-workers [50] suggested that this did not significantly contribute to the high heterogeneity observed in neonatal studies investigating the use of CD64. The source of this may have been differences in early- and late-onset neonatal sepsis, which, as alluded to previously, should be considered separate entities, each with an appropriate diagnostic approach.

The comprehensive meta-analysis of IL-6 measurement in critically ill adults conducted by Molano Franco and co-workers [104] revealed an extremely wide range of diagnostic cut-off values from 40 to 200,000pg/mL; it was again noted that no studies had pre-specified diagnostic cut-off values, which increased the risk of bias. The lack of pre-specified diagnostic cut-offs was also evident in neonatal studies [102], which was also compounded by differences in the classification of neonatal sepsis, which were both noted as limitations of their meta-analysis. Differing diagnostic cut-off values have also been suggested for sTREM-1, with a range of 1.25 to 525pg/mL found in the studies considered by Pontrelli and co-workers [105]. They further stated that there should be an effort to standardise analytical methods in order to enable the comparison of results between studies. The use of suPAR for both diagnosis and prognosis of sepsis has produced a range of cut-off values with the range for prognosis slightly higher than that for diagnosis (6.3 to 14.3ng/mL and 2.7 to 12ng/mL respectively) [108]. Once again these values have been calculated retrospectively, rather than being pre-specified.

It can be appreciated that there are a wide range of potential diagnostic cut-off values for all biomarkers described; the main limitation when attempting to investigate the possible diagnostic potential of these is the lack of pre-specified diagnostic cut-off values, which can in turn increase the risk of study bias and heterogeneity. The use of different methods may also be a contributing factor, although one approach to address this could be the use of recognised standards against which methods could be calibrated in an attempt to reduce variation.

## 7. Point-of-Care Testing (POCT) for Sepsis

The biomarkers currently measured for the investigation of sepsis are generally measured in the central laboratory environment. Such testing is usually performed on large, automated clinical chemistry analysers that employ a combination of ion-selective electrodes and colourimetric, immunoturbidimetric and various immunoassay techniques. The required specimen types are serum and plasma and are obtained by venepuncture in evacuated blood collection tubes. As such, there is an inherent delay in obtaining results due to sample transportation to the laboratory and pre-analytical steps such as centrifugation. Interest has therefore turned to point-of-care testing (POCT) devices that can be used to provide rapid results and reduce the time before appropriate treatment is started [113]. As alluded to previously, the only biomarkers that are currently included in the diagnostic workup of suspected sepsis are CRP and lactate. POCT devices for both of these biomarkers are currently available. A large evaluation study by Brouwer and van Pelt [114] compared eight POCT devices, all of which are commercially available, to a routinely used central laboratory CRP assay. Their work highlighted the variability in the precision of these methods and the comparability with the laboratory assay. Despite the varying assay performance noted, they did conclude that three of the assays evaluated could potentially be used in the point-of-care setting to provide reasonably reliable, precise and relatively accurate results. In the UK, NICE have produced innovation briefings for three CRP POCT methods, which describe the technology and review the available evidence regarding their use [115,116,117]. Interestingly, all three devices are suggested for use in the primary care setting, mainly to support the identification of potential lower respiratory tract infections which may be amenable to antibiotic therapy. This highlights an important factor that must be addressed when considering POCT; such testing must always be used in the appropriate clinical context, and the performance, limitations and cost (both in terms of outlay and potential cost savings) of the assay fully considered. Lactate may be measured on portable POCT devices and on blood-gas analysers equipped with a lactate-measuring electrode [118,119]. Blood-gas analysers are usually bench-top analysers that are generally not portable; however, they are routinely used in emergency departments, critical care environments and other hospital wards, as well as in central laboratories. Their placement in such locations clearly designated them as POCT devices, despite their lack of portability. Hand-held devices incorporate similar measuring technology as the larger blood-gas analysers for measurement of lactate (direct ion-selective electrodes), and their performance is comparable to that of blood-gas analysers [118,119].

POCT methods for PCT are also commercially available. Brahms originally produced the first automated monoclonal sandwich PCT assay, which is available for several clinical chemistry immunoassay analytical platforms; this was subsequently followed by the release of a POCT PCT assay (Brahms PCT Direct) which demonstrated very good performance, comparable to that of their original PCT assay [120]. Radiometer have also developed a POCT PCT assay for their AQT90 Flex analyser, which has also been shown to have good correlation with the original Brahms PCT assay. Although both the Brahms and Radiometer assays are described as POCT, there are inherent differences that should be recognised. The Brahms PCT direct assay uses a portable reader system and the required specimen for analysis can be either a capillary blood sample obtained via a finger prick or whole blood collected by venepuncture. The Radiometer ACT90 Flex PCT assay is performed on the AQT90 Flex analyser, which is a bench-top instrument similar in footprint to a blood-gas analyser; the required specimen is 2mL of whole blood or plasma, which therefore requires venepuncture. This again highlights the differences within the scope of POCT, which must be considered if such testing is to be implemented in routine practice.

The feasibility of POCT for sepsis-related biomarkers has clearly been demonstrated by the existing commercially available assays that are routinely used for biomarkers such as CRP, lactate and PCT. Research and development focusing on new methods and techniques with the aim of developing POCT devices is a highly active area [121]. Microfluidic devices that enable the miniaturisation of conventional analytical technologies to facilitate POCT is one area of interest, since it can allow the automation of complex fluidic processes [122]. There are, however, a number of issues that need to be solved, including cost, manufacturing scalability and analytical quality [123]. Nevertheless, progress has been made with the development of biosensors that can be incorporated into these devices for the measurement of biomarkers such as PCT [124]. Recent work has led to the development of a microelectrode sensor for the measurement of IL-6 with an assay time of around 2.5 min [125]. Work on more novel biomarkers has also been done by Min and co-workers [126], who developed a biosensor for the measurement of IL-3 and have attempted to validate its use in a small cohort of septic and control patients. In addition to biosensors, increasing the sensitivity of more widely used absorption spectrophotometric measurements has been achieved through the use of techniques such as cavity-enhanced absorption spectrometry (CEAS) [127]. This has a 30- to 100-fold increase in sensitivity compared to conventional single-pass absorbance detection. This high sensitivity is achieved by placing the sample within an optical cavity, formed with two highly reflective dielectric mirrors, where the light passes back and forth between the mirrors, thus increasing the absorption effect. Coupling of this technique with enzyme-linked immunosorbent assay (ELISA) has been achieved [128], demonstrating the feasibility of this approach. Optics comprised of interferometric biosensors coupled with immunoassay have been constructed, using antibodies to CRP and IL-6 which are linked directly to the sensor chip, thereby removing the requirement of a second, labelled antibody [129]. Although the concept in this study was to provide multiplex measurement of CRP and IL-6, the results suggested that there was some degree of cross reactivity and the authors concede that the use of a second antibody and reverting to the more traditional sandwich antibody format may be necessary. However, the concept of direct antibody measurement is still appealing and is worth further investigation. Moreover, ELISAs can be implemented in a variety of formats, as an example a simple dip strip ELISA [130], which could offer benefits in terms costs and performance.

More promising multiplexing work has been conducted by Molinero-Fernández and co-workers [131], who successfully developed an electrochemical magnetoimmunosensor for the simultaneous measurement of CRP and PCT, and demonstrated comparable performance to the Brahms CRP and PCT assays performed on the Brahms Kryptor analyser, in part due to the analysis of specimens from neonates with suspected sepsis. Although only three clinical samples were analysed, the results did indicate good agreement with the Brahms Kryptor assays and imprecision studies revealed that the reproducibility of measurements made by the electrochemical magnetoimmunosensor was very good (eight replicate measurements performed a day for five days). The CRP and PCT multiplexing technique requires 30µL of plasma and returns a result within 20 min. This is therefore a very promising approach, although the need for plasma instead of whole blood necessitates pre-analytical centrifugation. Lakey and co-workers reported an impedimetric electrode array within a polymer microfluidic cartridge that demonstrates an approach that could be used for multiplexed sensing of an array of sepsis biomarkers for low-cost point-of-care diagnostics [132].

One limitation of the measurement of CD64 is the requirement for flow cytometry, which is generally only available in specialised laboratories. Work has been done to incorporate a CD64 assay into a point-of-care microfluidic device using an immunoaffinity capture technique [133,134]. These are not true POCT devices yet, and have assay times of up to two hours, so they are not yet ready for routine use; however, as a proof-of-concept approach they do demonstrate potential. An approach adopted by Ghonge and co-workers [100] was to couple smartphone-based imaging with a microfluidic biochip to measure CD64. Whole blood passes through the biochip, which contains CD64 antibodies. The immobilised cells are then imaged with the smartphone and measurements from the image used to determine the number of CD64 neutrophils present in the blood sample. In addition to demonstrating the feasibility of their technique, comparisons to a flow cytometry method were made through the measurement of 37 blood samples, which indicated that there was correlation between the two methods (R^2^ = 0.88).

## 8. Identification of New Biomarkers and Diagnostic Tools

The potential for the discovery of new biomarkers has increased through the use of proteomic and metabolomic techniques, particularly due to the advances made in mass spectrometry [135]. This approach has already been used to identify a panel of metabolites, described as a “biopattern”, for the triage of paediatric patients with suspected sepsis [136]. The authors suggested that the metabolites are related to pathways involved in energy metabolism and are indicative of the changes induced in these systems during sepsis. Although the analytes identified are generally not routinely analysed, this approach shows promise as it may be able to identify specific biomarkers that are consistently altered in keeping with the pathobiology of sepsis. These “omics” approaches are evolving areas of development and, while possibly being some way from providing approaches that could be adapted to routine analysis, they may eventually provide more powerful approaches. The biomarkers or biopatterns identified through the use of these techniques are based on a patient’s genotype and phenotype, which could produce more disease- or condition-specific signatures for more accurate diagnosis [38].

Another class of potential biomarkers is that of non-coding RNAs, particularly microRNAs, although these are very much still a research-based application at present [137]. MicroRNAs are involved in the regulation of gene expression by inhibiting the translation of messenger RNA [138]. Certain microRNAs have been associated with sepsis; a study by Han and co-workers [139] demonstrated an elevation of microRNA-143 in septic patients, which was significantly higher than either patients classified using the SIRS criteria as suffering from SIRS, or healthy controls. They also noted that microRNA-143 correlated with SOFA, although it did not predict mortality. Guo and co-workers [140] measured microRNA-495 in 103 patients diagnosed with sepsis. They found that microRNA-495 was reduced in patients with sepsis, and was further reduced in patients identified as suffering from septic shock. They suggested that microRNA-495 is a possible biomarker for sepsis, and that it can also potentially discriminate between patients with sepsis and those who have progressed to septic shock. A more complex approach was attempted by Caserta and co-workers [141], who used next-generation sequencing to establish a panel of candidate microRNAs that were altered between critical care patients with sepsis, non-infective SIRS and no SIRS. They validated this panel and identified six microRNAs that were reduced in sepsis and SIRS, enabling them to distinguish between the three patient groups with relatively high accuracy. Interestingly, they noted that the levels of these were inversely correlated with pro-inflammatory-associated biomarkers such as IL-6 and CRP. As with omics-based approaches, while microRNAs are perhaps not yet ready for widespread routine use, these studies indicate that they may be useful tools in the future.

Another approach to sepsis diagnosis has been the development of computer-based predictive algorithms. Calvert and co-workers [142] developed the “InSight” algorithm, which they claim can predict the onset of sepsis three hours before the manifestation of a sustained SIRS episode. This predictive ability was calculated to have an AUC of 0.92, which they claimed outperforms most existing biomarkers. The algorithm itself is based on measurement of age, heart rate, oxygen saturation, pH, pulse pressure, respiration rate, temperature, systolic blood pressure and white cell count; these were chosen as they are standard, routinely monitored parameters not only in patients with suspected sepsis, but most patients in general. This type of approach could be attractive due to the use of routinely measured parameters without the need for any extra testing. Further work by Calvert and colleagues [143] narrowed their focus to develop a machine-learning algorithm based on a patient’s age and length of stay in hospital. They purposely chose a high-risk patient group and noted that this algorithm is only applicable to this specific group. However, their work indicated that applying the algorithm within an hour of a patient becoming symptomatic significantly outperformed the existing scoring systems for prediction of sepsis. The application of machine-learning algorithms with the advent of “big data”, and the possible combination with existing and emerging biomarkers may provide more focused diagnostic strategies if properly applied.

## 9. Conclusions

Sepsis is clearly an important syndrome associated with significant morbidity and mortality, the true extent of which is not fully understood due to its variability and the lack of specific epidemiological data. The mechanisms that underlie the pathobiology of sepsis are now better understood, which has resulted in the definition of sepsis being re-revaluated and redefined, although much is still to be elucidated. The biomarkers associated with sepsis are linked to the pathways and processes that are part of this pathobiology. It is therefore unsurprising that they have become the focus of much research and study.

The studies presented herein highlighted some key points regarding the potential use of biomarkers for sepsis, and identified candidate biomarkers that have been the focus of much of the recent research (see Table 6). Various studies have suggested differing diagnostic accuracies for these biomarkers; smaller scale studies tend to demonstrate greater diagnostic power than larger studies. This is perhaps not surprising as the smaller studies have generally been conducted in specific patient cohorts in which the pre-test probability of a patient suffering from sepsis is higher than in the general patient population. It is important to consider which definition of sepsis has been used in these studies. Although the currently accepted definition of sepsis as defined by the Sepsis-3 taskforce was introduced in 2016, many studies still refer to older definitions, which can alter the diagnostic definitions of these biomarkers. Ideally, all studies should use the Sepsis-3 definitions from now on. Despite this, the general consensus is that these biomarkers do at least aid the prediction or diagnosis of sepsis. Another key point is that no one individual biomarker has been found to have sufficient diagnostic power to be used as a stand-alone diagnostic tool; however, their usefulness is increased by using a panel of biomarkers, which increases diagnostic accuracy above that of the individual biomarkers. This approach seems sensible; due to the complexity of sepsis it is unlikely that one single biomarker, unless positioned at some as yet undiscovered single critical point in the pathobiology of sepsis (which is unlikely given the multiple pathways and systems involved) would be able to provide definitive diagnostic information. Combining biomarkers that reflect different aspects of sepsis pathobiology potentially provides more useful information; however, more studies are probably needed to determine which biomarkers or combinations of biomarkers provide the most useful diagnostic information for the investigation of a patient with suspected sepsis.

Further consideration needs to be given to the approaches used to measure these biomarkers. If they are to be incorporated into a panel, then the same analytical technique may be required, especially if the ultimate aim is to enable their use for POCT. The low concentrations of some of these biomarkers would require analysis that is suitably sensitive and specific. POCT is an attractive option, especially when considering its potential use in the setting of sepsis. Early diagnosis is key so that appropriate therapy can be instigated as soon as possible. The more recent developments have demonstrated that current technology may enable the construction of multiplexing POCT devices that are able to measure a specific panel of appropriate biomarkers with rapid time to result from a very small sample, possibly whole blood, which is easily obtainable from a finger prick sample. Such devices could be a valuable addition to the sepsis care pathways if deployed in the appropriate clinical areas so that patients with sepsis can be identified at the earliest opportunity.

## Figures and Tables

**Figure 1 micromachines-11-00286-f001:**
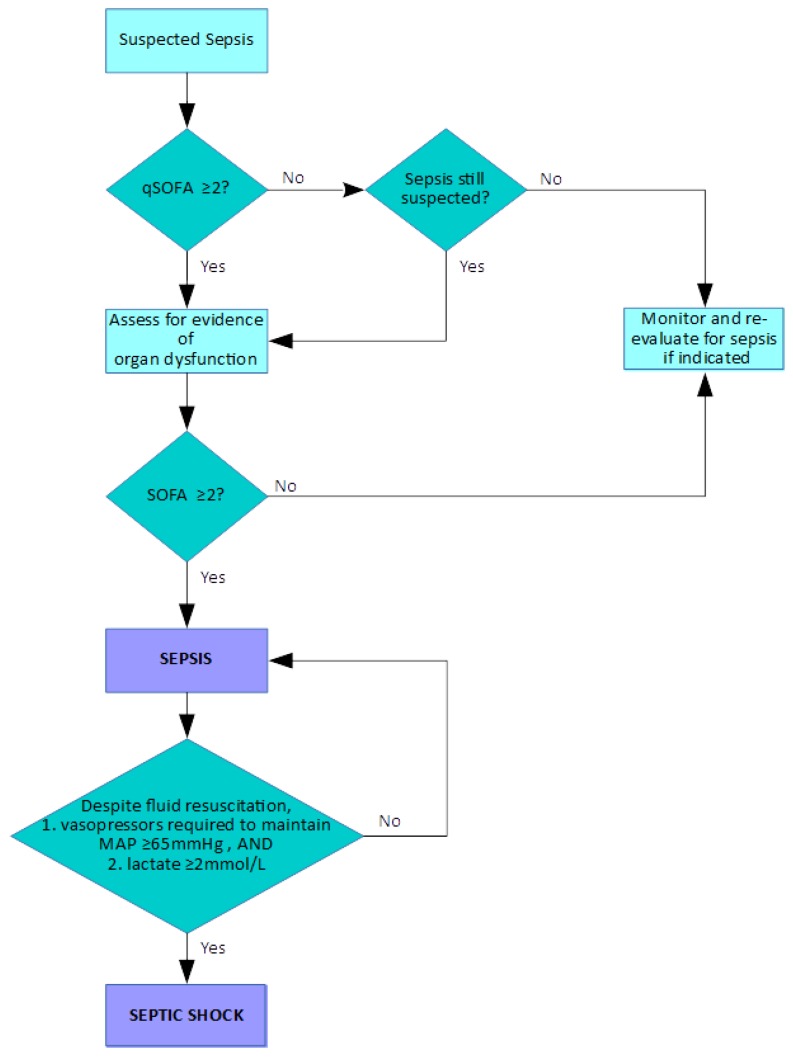
Clinical criteria for identifying sepsis and septic shock, adapted from Singer et al. [22]. qSOFA: Quick Sequential Organ Failure Assessment; SOFA: Sequential Organ Failure Assessment.

**Figure 2 micromachines-11-00286-f002:**
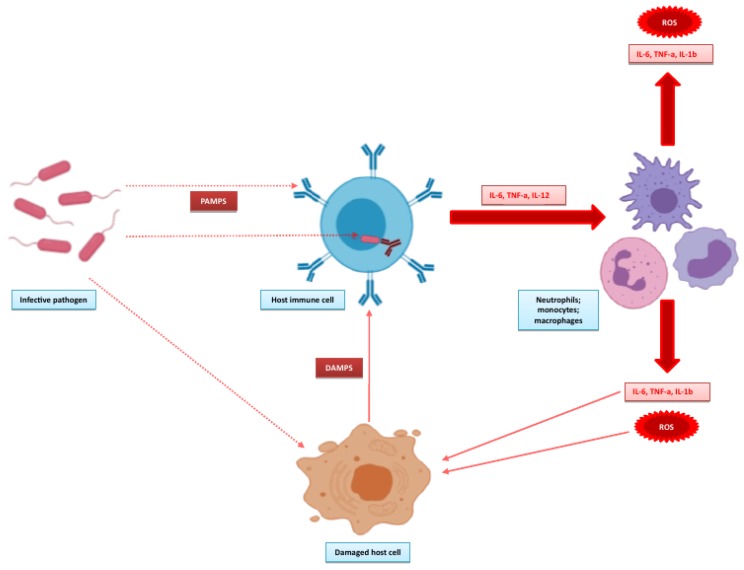
Pro-inflammatory cascade in sepsis (adapted from the works of Angus and van der Poll [30] and Wiersinga and co-workers [31]. Infective agents release pattern associated molecular patterns (PAMPS) which are recognised by host immune cells via pattern recognition receptors both extra- and intracellularly. These cells also recognise damage-associated molecular patterns (DAMPS) released by damaged host cells. In response, the immune cells release inflammatory cytokines, activating leucocytes which in turn release cytokines and reactive oxygen species, further promoting the inflammatory response, which can affect host cells as well as the invading pathogen.

**Figure 3 micromachines-11-00286-f003:**
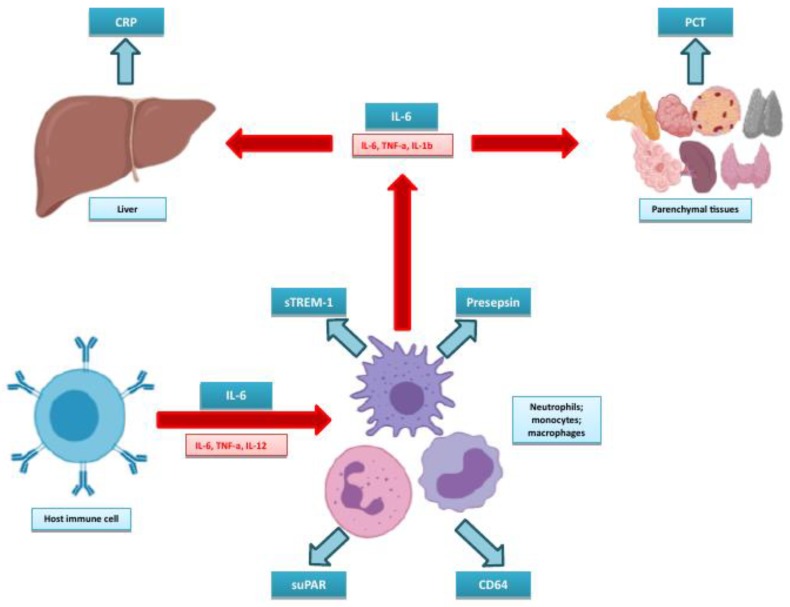
The origins of sepsis biomarkers in relation to the pro-inflammatory cascade activated during sepsis. CRP: C-reactive protein; CD64: cluster of differentiation 64; IL-6: interleukin-6; PCT: procalcitonin; sTREM-1: soluble triggering receptor expressed by myeloid cells-1.

**Table 1 micromachines-11-00286-t001:** Original definitions of sepsis and related terms, adapted from Bone et al. [20].

Systemic Inflammatory Response Syndrome	Sepsis	Severe Sepsis	Septic Shock
Systemic inflammatory response that occurs regardless of cause, i.e., infective or non-infective.	Sepsis is the systemic inflammatory response to infection.	Severe sepsis is sepsis associated with organ dysfunction, hypoperfusion abnormality or sepsis-induced hypotension.	Septic shock is a subset of severe sepsis and is defined as sepsis-induced hypotension, persisting despite adequate fluid resuscitation, along with the presence of hypoperfusion abnormalities or organ dysfunction.

**Table 2 micromachines-11-00286-t002:** Updated 2001 SIRS diagnostic criteria adapted from Levy et al. [21].

General Parameters	Inflammatory Pararmeters	Haemodynamic Parameters	Organ Dysfunction Parameters	Tissue Perfusion Parameters
Fever	Leukocytosis	Arterial hypotension or decreased systolic blood pressure	Arterial hypoxaemia	Hyperlactaemia
Hypothermia	Leukopaenia	Mixed venous oxygen saturation >70%	Acute oliguira	Deceased capillary refill or mottling
Heart rate > 90bpm	Normal white cell count with >10% immature forms	Raised cardiac index	Increased creatinine	-
Tachypnoea > 30bpm	CRP raised >2SD above normal	-	Coagulation abnormalities	-
Altered mental status	Procalcitonion >2SD above normal	-	Ileus	-
Significant oedema or positive fluid balance	-	-	Thrombocytopaenia	-
Hyperglycaemia	-	-	Hyperbilirubinaemia	-

**Table 3 micromachines-11-00286-t003:** “Sepsis-3” definitions and identifying features, adapted from Singer et al. [22].

Sepsis	Septic Shock
Sepsis is a life-threatening organ dysfunction caused by a dysregulated host response to infection.	Septic shock is a subset of sepsis in which underlying circulatory and cellular metabolism abnormalities are profound enough to substantially increase mortality.
Organ dysfunction can be identified as an acute change in total SOFA score greater or equal to 2 points consequent to the infection.	Patients with septic shock can be identified with a clinical construct of sepsis with persisting hypotension requiring vasopressors and having a serum lactate level > 2 mmol/L despite adequate volume resuscitation.

**Table 4 micromachines-11-00286-t004:** SOFA scoring system, adapted from Singer et al., 2016 [22].

System	Score				
	0	1	2	3	4
Respiration; PaO_2_/FiO_2_ (kPa)	≥53.3	<53.3	<40	<26.7 with respiratory support	<26.7 with respiratory support
Coagulation; Platelets	≥150	<150	<100	<50	<20
Liver; Bilirubin (µmol/L)	<20	20–32	33–101	102–204	≥204
Cardiovascular	MAP ≥70mmHg	MAP <70mmHg	Dopamine <5 or dobutamine administration	Dopamine 5.1–15 or epinephrine ≤0.1 or norepinephrine ≤0.1	Dopamine >15 or epinephrine >0.1 or norepinephrine >0.1
Central Nervous System; Glasgow Coma Score	15	13–14	10/12/19	06/09/19	<6
Renal; Creatinine (µmol/L)	<110	110–170	171–299	300–440	>440
Renal; Urine output (mL/day)	-	-	-	<500	<500

PaO_2_/FiO_2_: Fraction of inspired oxygen/partial pressure of oxygen; MAP: mean arterial pressure. Note: Catecholamine doses are µg/kg/min administered for at least 1 h.

**Table 5 micromachines-11-00286-t005:** Summary of most studied biomarkers identified by Liu and co-workers. The number of studies found, total number of patients from all studies for each biomarker, and the overall diagnostic accuracy (area under the curve, AUC) for each for the diagnosis of sepsis are shown. Adapted from Liu et al., 2016 [40].

Biomarker	Number of Studies	Number of Patients	Diagnostic Accuracy (AUC)
Procalcitonin	59	7376	0.85
CRP	45	5654	0.77
IL-6	22	3450	0.79
Soluble triggering receptor expressed on myeloid cells-1 (sTREM-1)	8	831	0.85
Presepsin	9	1510	0.88
Liposaccharide binding protein	5	1136	0.71
CD64	4	558	0.96

**Table 6 micromachines-11-00286-t006:** Summary of the nine biomarkers most studied for sepsis diagnosis, as identified by the current literature search.

Biomarker	Number of Studies	Number of Meta-Analyses	Study Population and Setting			
			Neonates	Adults, ED	Adults, ICU	Other
Procalcitonin	27	2	7	2	17	Adults, various wards = 2; Neonates and paediatrics, various wards = 1
Presepsin	19	4	11	2	8	Adults, various wards = 1; Paediatrics, various wards = 1
CD64	14	3	6	3	7	Adults, various wards = 1
CRP	12	0	6	1	5	0
IL-6	9	2	5	0	5	Non-clinical, experimental study = 1
sTREM-1	6	1	1	0	5	Neonates and paediatrics, various wards = 1
Lactate	4	0	0	2	1	Various patients, various healthcare settings = 1
Neutrophil to lymphocyte count ratio	4	0	0	1	2	Febrile paediatrics, aged between 1 month and 5 years = 1
suPAR	3	1	0	0	3	Adults, ED or ICU

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
