# Peer review of "Biomarkers for Point-of-Care Diagnosis of Sepsis"

_micromachines, 2020, doi:10.3390/mi11030286_

Round 1

Reviewer 1 Report

In the present manuscript, Teggert et al. discussed current sepsis management, diagnosis, and biomarkers. Sepsis is a life-threatening disease that could develop rapidly. Hence, point of care testing is especially valuable for sepsis. Authors provided an excellent summary on the definition of sepsis as sepsis is a complex disease itself, which is useful for the readers of Micromachines who are technology experts. The list of biomarkers is also valuable and up to date. I recommend the publication of this review after addressing the following concerns.

Authors should provide clinically relevant concentration and their change in normal and disease states for each biomarker. Such information is important for developing point of care testing methods. Developing point-of-care testing is always is trade-off between simplicity and accuracy/sensitivity. Knowing the target concentration will help refine the method development efforts. Section 7 “Point-of-care testing (POCT) for Sepsis” should be expanded. More detailed discussion on each biomarker and their suitability/difficulty for POCT should be provided.

Author Response

The valid points raised by the reviewer have been addressed in the updated revision.  An extra section has been included which details the current issues with regards to reference limits and diagnostic cut off values for each of the biomarkers.

Reviewer 2 Report

The manuscript is well written and organized. 

Author Response

Thank you for your review.  Revisions have been made to the original document.